# Claudin 18.2 as a New Biomarker in Gastric Cancer—What Should We Know?

**DOI:** 10.3390/cancers16030679

**Published:** 2024-02-05

**Authors:** Maria Cecília Mathias-Machado, Victor Hugo Fonseca de Jesus, Alexandre Jácome, Mauro Daniel Donadio, Marcelo Porfirio Sunagua Aruquipa, João Fogacci, Renato Guerino Cunha, Leonard Medeiros da Silva, Renata D’Alpino Peixoto

**Affiliations:** 1Division of Gastrointestinal Medical Oncology, Oncoclínicas, São Paulo 04538-132, Brazil; maurodsd@gmail.com (M.D.D.); marcelo.aruquipa@medicos.oncoclinicas.com (M.P.S.A.); renatadalpino@gmail.com (R.D.P.); 2Division of Gastrointestinal Medical Oncology, Oncoclínicas, Florianópolis 88015-020, Brazil; victor.jesus@medicos.oncoclinicas.com; 3Division of Gastrointestinal Medical Oncology, Oncoclínicas, Belo Horizonte 30360-680, Brazil; alexandre.jacome@medicos.oncoclinicas.com; 4Division of Gastrointestinal Medical Oncology, Oncoclínicas, Rio de Janeiro 22775-003, Brazil; joao.fogacci@medicos.oncoclinicas.com; 5Cellular Therapy Program, Division of Hematology, Oncoclínicas, São Paulo 04538-132, Brazil; renato.cunha@medicos.oncoclinicas.com; 6Oncoclinicas Prescision Medicine, Oncoclínicas, São Paulo 04513-020, Brazil; leonard.silva@ocpmedicine.com

**Keywords:** cancer, gastric cancer, treatment, biomarkers

## Abstract

**Simple Summary:**

Gastric cancer is a major global health issue, ranking among the top causes of cancer-related deaths. Unfortunately, most patients are diagnosed in advanced stages, with only a 6% chance of surviving five years. Current treatment involves a specific drug combination, but predicting its effectiveness is challenging. This review explores the changing world of gastric cancer markers, focusing on Claudin 18.2 (CLDN18.2) as a promising target. Recent trials show that Zolbetuximab, a CLDN18.2-targeting drug, works well with chemotherapy for CLDN18.2-positive gastric cancer. Understanding how to choose these therapies is crucial as this new approach unfolds. This review aims to overview new strategies in managing CLDN18.2 positive advanced gastric cancer.

**Abstract:**

Gastric cancer (GC) remains a formidable global health challenge, ranking among the top-five causes of cancer-related deaths worldwide. The majority of patients face advanced stages at diagnosis, with a mere 6% five-year survival rate. First-line treatment for metastatic GC typically involves a fluoropyrimidine and platinum agent combination; yet, predictive molecular markers have proven elusive. This review navigates the evolving landscape of GC biomarkers, with a specific focus on Claudin 18.2 (CLDN18.2) as an emerging and promising target. Recent phase III trials have unveiled the efficacy of Zolbetuximab, a CLDN18.2-targeting antibody, in combination with oxaliplatin-based chemotherapy for CLDN18.2-positive metastatic GC. As this novel therapeutic avenue unfolds, understanding the nuanced decision making regarding the selection of anti-CLDN18.2 therapies over other targeted agents in metastatic GC becomes crucial. This manuscript reviews the evolving role of CLDN18.2 as a biomarker in GC and explores the current status of CLDN18.2-targeting agents in clinical development. The aim is to provide concise insights into the potential of CLDN18.2 as a therapeutic target and guide future clinical decisions in the management of metastatic GC.

## 1. Introduction

Gastric cancer (GC) is an important global health-care challenge and accounts for one of the top-five leading causes of cancer-related deaths worldwide, with 768,793 deaths estimated in 2020 [1]. Unfortunately, most patients are diagnosed at advanced stages with a five-year survival rate of only 6% [2,3].

A combination of a fluoropyrimidine (capecitabine or 5-fluorouracil) plus a platinum agent (oxaliplatin or cisplatin) has remained as the first-line choice of treatment for most patients with metastatic GC over the past few decades. The identification of predictive and actionable molecular abnormalities has been challenging in GC. Epidermal growth factor (EGF), MET, and vascular endothelial growth factor (VEGF) are examples of potential therapeutic targets that have been proved unsuccessful in first-line strategies [4,5,6,7,8,9]. However, molecularly driven approaches have been gaining attention since 2010. HER-2 was the first biomarker to allow the incorporation of an effective targeted therapy—trastuzumab—in combination with chemotherapy in the first-line setting for patients with an immunohistochemistry (IHC) score of 3+ or with FISH positivity and an IHC score of 2+ [10]. Indeed, higher HER2 expression and amplification have been linked to a greater benefit from treatment regimens containing trastuzumab [10].

Later, both PD-L1 expression, measured using the combined positive score (CPS), and microsatellite instability-high (MSI-H)/mismatch repair deficiency (MMR-D) status were established as predictive biomarkers of benefit for anti-PD1 monoclonal antibody therapies in combination with chemotherapy for first-line treatment of metastatic GC [11,12]. The addition of both nivolumab and pembrolizumab has proved to increase overall survival (OS) when compared to chemotherapy alone in the phase III trials Checkmate-649 and Keynote-859, respectively, with the clinical benefit, however, being limited to those with higher CPS scores [11,12]. Nonetheless, the ideal cutoff value of the CPS to truly differentiate tumors that derive or not benefit from immunotherapy is yet to be determined.

More recently, Claudin 18.2 (CLND18.2) entered this select group of biomarkers in GC. Zolbetuximab, a chimeric immunoglobulin G1 antibody specific for CLND18.2, has recently demonstrated benefit in combination with oxaliplatin-based chemotherapy for CLND18.2-positive metastatic GC based on the results of two large phase III trials [13,14]. With the advent of this novel therapeutic strategy, it becomes necessary to understand when anti-CLND18.2 therapies should be chosen over or in combination with other targeted agents in metastatic GC. The aim of this manuscript is to review the role of CLDN18.2 as a biomarker in GC and to explore the status of CLDN18.2-targeting agents in clinical development.

## 2. Claudins

Cancer metastasis requires local infiltration into the adjacent stroma and surrounding normal cells. As such, the epithelial–mesenchymal transition (EMT) plays a valuable role in tissue invasion by enabling tumor cells to assume a mesenchymal phenotype leading to cell migration, invasiveness, and resistance to apoptosis [15]. For such processes to initiate, there is a progressive loss of epithelial cell markers which results in an increased cellular migration capacity and, therefore, higher metastatic potential. Contact between epithelial cells is mediated by structures such as tight junctions (TJs), desmosomes, and gap and adherens junctions. The loss of the binding characteristics of these structures consolidates the first step in the EMT process.

TJs are endogenous proteins located at the cellular membrane composed, amongst others, of claudins (CLDNs)—an important group of 27 [16,17] proteins that act in epithelial cell layer permeability and polarity as well as cellular migration [18,19,20,21,22,23,24]. An analysis of claudin proteins has proposed the subdivision of the claudin family into ‘classic’ and ‘non-classic’ groups, according to the alternative splicing of the CLDN18 gene exons [25,26], and the classic (CLDNs 1–10, CLDN14, CLDN15, CLDN17, CLDN19) and non-classic types (CLDN 11–13, CLDN16, CLDN18, and CLDNs 20–24). However, more recently, claudins have been divided based on sequence homology [26]. Their structure is composed of N-terminal and C-terminal regions in the cytoplasm, two extracellular loops, and four transmembrane domains [26,27] (Figure 1). 

CLDNs are located at the apical region of the cell membrane and act in the TJ complex, promoting cell adhesion, maintaining cell polarity, and allowing cell migration, matrix remodeling, as well as cell proliferation. They also play an important role in selective paracellular permeability, which is pivotal as a barrier at the epithelial cellular layer [19,20,21,24,28,29]. The lack of CLDN expression may be associated with increasing levels of metastatic tumor cell migration and infiltration. On the other hand, CLDNs are commonly overexpressed in several cancers including GC [27], therefore being associated with a higher metastatic potential [27]. 

In normal gastric mucosa, CLDN1–5, CLDN7–12, CLDN16, and CLDN18 are constitutively expressed [30]. Also, several different CLDN family members were found to be expressed in GC [22]. However, CLDN18 has gained the most attention as a therapeutic target and has been more intensively studied in GC due to its normal expression in gastric cells combined with its consistent and stable expression in gastric tumor cells [31]. The *CLDN18* gene has two splice variants encoding two isoforms: CLDN18.1, expressed in normal and lung cancer cells, and CLDN18.2, which is almost exclusively present in normal gastric mucosa cells, although it may be expressed in lung, esophageal, pancreatic, and gastric tumor cells [29,32]. In differentiated and stem gastric epithelial cells, CLDN18.2 is present in the TJs [25,32,33,34] and regulates the permeability to both Na+ and H+ in gastric acid by functioning as a barrier in the gastric mucosa [35], playing a role in cellular polarity, retaining a barrier function, as well as promoting resistance to gastric acid [36]. 

Due to the location of CLDN18.2, during the malignant transformation of the normal epithelial cells of the gastric mucosa into GC, alterations in cell polarity lead to the exposure of the cellular surface that contains CLDN18.2 epitopes, functioning as a possible target for therapies such as monoclonal antibodies [32,37]. Also, CLDN18 expression is found to be different depending on the histological type of GC, with CLDN18.2 presenting higher expressions in diffuse GC [38,39]. 

Specifically in GC, CLDN18 appears to have different functions based on the level of expression in tumor cells. While highly expressed in normal gastric tissue, the downregulation of CLDN18 expression has been identified in approximately 58% of GCs, increasing to 74% in intestinal phenotype GC [40]. Also, CLDN18 downregulation is more frequently found in GC cells than in the surrounding gastric and intestinal metaplasia mucosa [41,42]. This feature may correlate with GC development, tumor proliferation, and infiltration. Interestingly, upon tissue analysis of early-stage GC, the Ki67 proliferation index at the invasive front seems to inversely correlate with the expression of CLDN18. Those findings suggest that both high proliferation and cancer cell invasion require a decrease in CLDN18 expression [42]. These finding can also raise the alarm for a possible correlation between CLDN18.2 expression and GC prognosis with studies suggesting that patients whose tumors maintain CLDN18 expression present a longer OS compared to those lacking CLDN18 expression [40,41].

It is also important to note that, while CLDN18 downregulation plays a part in GC proliferation, higher levels of CLDN18 expression are also of upmost importance. Correlative studies analyzing CLDN18.2 immunohistochemical expression in GC have been conducted with interesting findings. In one of those studies, 42.2% of GC (203 of 481) presented with a strong IHC staining (IHC 3+ score) and demonstrated a correlation with mucin phenotype and EBV status. However, no associations between CLDN18.2 expression and survival and the intestinal phenotype of other clinicopathological features were found [43]. In another Korean study, the clinical relevance of the expression of CLDN18.2 was evaluated in 367 GC patients. The authors reported 74.4% and 29.4% of positive expression, presenting with moderate-to-strong staining, respectively [44]. These positive expression rates were higher in patients with diffuse-type GC and HER2-positive tumors, but no correlation with survival or other characteristics was shown [44]. 

In addition, a study evaluating Japanese GC patients demonstrated CLDN18.2 positivity in 87% (228 of 262) of all primary tumors and 80% (108 of 135) of lymph node metastatic disease [38]. Also, upon analyzing CLDN18.2, a moderate-to-strong expression (CLDN18.2 expression ≥2+ membrane staining intensity in ≥40% of tumor cells) was encountered in 52% (135 of 262) of primary gastric tumors and 45% (61 of 135) of lymph node metastases [38]. The investigators also demonstrated a higher CLDN18.2 expression in Lauren-diffuse-subtype and high-grade tumors [38]. Regarding metastatic GC, CLDN18 expression has been demonstrated to be lower in patients with peritoneal and liver metastasis, but higher expression levels were observed in patients with bone and lymph node metastasis [45,46]. 

Data regarding the prevalence of CLDN18.2 expression throughout different populations around the globe are limited. However, a recently presented study reported the prevalence and biomarker analysis of over 4000 locally advanced unresectable or metastatic GC tumor samples tested for CLDN18.2 status from two different studies [47]. In this study, CLDN18 positivity was defined as ≥75% of tumor cells demonstrating a moderate-to-strong IHC staining, and 4507 patients presented with a valid CLDN18 IHC results, of which 1730 (38.4%) were CLDN18.2-positive tumors. As with other cited studies, a statistically significant association between CLDN18.2 status and tumor type was demonstrated (*p* = 0.0002): diffuse tumors presented the highest prevalence of CLDN18.2 positivity (48.3%) when compared to intestinal and mixed-type tumors (38.8% and 42.9%, respectively). In this study [47], significant correlations between CLDN18.2 expression and sex (*p* < 0.0001) were identified, with CLDN18.2 prevalence being higher in female patients (42.8%) compared to male patients (36.2%). Also, statistically relevant correlations were present between CLDN18.2 status and race (*p* = 0.0004), with CLDN18.2 prevalence being higher in White patients (42.3%) when compared to Asian patients (36.4%), raising the question of a possible specific population that may benefit more from anti-CLDN18.2 targeted therapies. Interestingly, this report demonstrated an accurate correlation of CLDN18.2 positivity from samples collected through biopsy (38.6%) and those collected through tumor resection (37.6%), as well as samples collected from metastatic sites (39.4%) and samples collected from primary tumor sites (38.0%). Across the globe, CLDN18.2 prevalence ranged from 30% to 44% across patients from all regions (Asia Pacific, China Mainland, North America, South America, and Europe/the Middle East). The prevalence of CLDN18.2-positive patient samples was 36.5% in the Asia Pacific region, 37.7% in North America, 35.0% in mainland China, 30.0% in South America, and 44.0% in Europe/the Middle East [47]. 

## 3. How to Test

The FDA has given priority review to the biologics license application (BLA) aiming for the approval of zolbetuximab (IMAB362). This is intended for the initial treatment of patients dealing with unresectable, locally advanced, or metastatic HER2-negative gastric or gastroesophageal junction adenocarcinoma, specifically those with CLDN18.2-positive tumors [48]. Supporting evidence for the BLA is derived from data gathered in the phase 3 SPOTLIGHT (NCT03504397) and GLOW (NCT03653507) trials, which will be further explored. Both the SPOTLIGHT (NCT03504397) and GLOW (NCT03653507) trials used immunohistochemistry (IHC) assays for CLDN18.2 detection and defined positivity as moderate-to-strong CLDN18.2 membrane staining in at least 75% of tumor cells [13,14].

There is a heterogeneity in terms of the methodology used for CLDN18 expression detection in the scientific literature. Currently, the only approved compendium diagnostic IHC test for CLDN18 detection is the VENTANA CLDN18 (43-14A) assay which is designed to immunohistochemically detect the CLDN18 protein in formalin-fixed, paraffin-embedded neoplastic tissues using BenchMark IHC/ISH instruments. This assay targets the C terminus of claudin 18, but it does not specifically differentiate isoform 18.2. On the other hand, several CLDN18 immunohistochemical detection assays were used in previous scientific publications. For instance, the anti-CLDN EPR19202 kit (Abcam, Cambridge, UK), which is specific for the CLDN 18.2 isoform, was used in Leica Bond-Max Autostainer for CLDN18 expression detection of esophageal and gastric cancer [43,49].

Although the SPOTLIGHT and GLOW trials defined positivity as moderate-to-strong CLDN18.2 membrane staining in at least 75% of tumor cells [13,14], the scoring cut-offs and methodology used for CLND18 expression evaluation vary in previous publications, reaching values as low as 5% for the number of positive cells used to score a sample as positive and the scoring methods using H-score and IRSs [38,43,44,49,50,51]. One shall abide by the cut-offs of diagnostic assays and cut-offs used in the studies which granted approval for the use of zolbetuximab [13,14]. Future studies may shed light on how different scoring systems, antibody clones, and cut-offs can correlate with clinical and pathological parameters and, therefore, allow even better treatment selection.

## 4. Prognostic Value of CLDN18.2

Several studies have reported putative associations between CLDN18.2 expression and tumor extent and survival in GEJ/GC [41,43,44,50,52,53,54,55,56,57,58,59]. However, most of the observed clinical data are correlative and derived almost exclusively from retrospective studies. Those studies have used different criteria of positivity for CLDN 18.2 expression or assessment methods and have even jointly analyzed the disease at different stages. Furthermore, some of these studies provide contradictory data for the same CLDN molecules, suggesting that the expression and functions of CLDN18.2 may vary according to the stage of the disease, hindering their applicability as prognostic markers (Table 1).

Although some studies suggest that a greater expression of CLDN-4, -6, and -7 is correlated with shorter OS [57,59,60], others point towards no prognostic impact [43,44,50,52,53,54,55,56]. A meta-analysis comprising nine studies involving 1265 gastric cancer (GC) patients indicated that CLDN-4 overexpression correlated with poorer overall survival (OS) (95% CI 1.62–2.50; odds ratio (OR) 2.01), advanced stage (95% CI 1.08–3.56; OR 1.96), and lymph node metastasis (95% CI 1.05–2.81; OR 1.72). However, no significant association with distant metastasis was observed. Despite efforts, the meta-analysis could not eliminate the heterogeneity across studies, stemming from diverse immunohistochemical (IHC) methods, such as variations in the primary antibody, antibody dilutions, and the scoring system applied, potentially influencing the collective findings [61]. More recently, yet another meta-analysis, comprising six studies and involving 2440 gastric cancer (GC) patients, provided a well-defined scoring system for CLDN18.2 staining. The analysis categorized studies into two subgroups based on the definition of CLDN18.2 positivity: either staining intensity present in any percentage of tumor cells or staining intensity present in more than 40% of tumor cells. Interestingly, this meta-analysis did not uncover significant correlations between CLDN18.2 expression and TNM stages, the Lauren classification, HER2 status, or tumor grading. Still, no significant difference in OS was found between the groups of CLDN18.2-positive and -negative tumors (95% CI 0.69–1.48; HR 1.01; *p* = 0.95) [62].

It is crucial to note that two additional studies, which were not part of the earlier meta-analysis, examined the prognostic role of CLDN18.2 using the current positivity criteria (moderate-to-strong expression in ≥75% of tumor cells). In the study by Pellino and colleagues, CLDN positivity showed no correlation with overall survival (OS) in advanced gastroesophageal junction (GEJ)/gastric cancer (GC), although this study encompassed stages I–III. The expression of CLDN18.2 at various cut-offs (<50%; ≥50%; ≥75%) did not demonstrate a link to either shortened or prolonged OS time (*p* = 0.9264) [54]. Similarly, in the study by Kubota and colleagues, there was no significant difference in median progression-free survival (PFS) and OS between CLDN-positive and -negative patients who received standard first-line treatment: 8.6 vs. 7.1 months (95% CI 0.73–1.43; HR 1.02; *p* = 0.895) and 18.4 vs. 20.1 months (95% CI 0.89–1.78; HR 1.26; *p* = 0.191), respectively. Also, the efficacy of second-line chemotherapy did not differ between CLDN-positive and -negative groups: median PFS 4.2 vs. 4.0 months (95% CI 0.80–1.44; HR 1.08; *p* = 0.625) [56]. 

Therefore, by combining the evidence to date, the prognostic role of CLDN18.2 appears to have no clinical relevance. Its predictive impact, however, is more well established and grounds the most recent advances in metastatic GC therapy.

## 5. Predictive Value of CLDN 18.2

As previously mentioned, the cut-off value for CLDN18.2 expression used in clinical trials to predict the efficacy of currently available therapies is a sensitive point of discussion. In the phase IIa MONO trial, which evaluated the use of zolbetuximab monotherapy in late-stage advanced gastroesophageal cancer, a standardized IHC protocol was employed using an anti-claudin 18.2 [43-14A] rabbit antiserum [CLAUDETECT 18.2] to define the following: absence of staining (0), weak staining (1+), moderate staining (2+), and strong staining (3+). In this study, only patients with moderate and strong staining (2+ or 3+), with membrane staining intensity in ≥50% of tumor cells, were considered eligible. Out of the 43 evaluable patients, the overall response rate (ORR) was 9% (n = 4), with a clinical benefit rate of 23% (n = 10). Remarkably, 9 out of these 10 patients exhibited moderate-to-strong expression in ≥70% of tumor cells [63].

The phase II FAST trial randomized patients with advanced gastroesophageal cancer in first-line palliative settings to chemotherapy alone, zolbetuximab monotherapy, or chemotherapy combined with zolbetuximab. The IHC criteria for CLDN positivity involved 2+ or 3+ staining in ≥40% of tumor cells. Median PFS (7.5 versus 5.3 months) and OS (13 versus 8 months) favored the combination arm when compared to chemotherapy alone, corresponding to risk reductions of 56% for progression and 45% for death. Upon analyzing the 2+ or 3+ staining in ≥70% of tumor cells, an enhanced clinical benefit was noted, with a median PFS of 9.0 months (versus 5.7 months) and a median OS of 16.5 months (versus 8.9 months). Within the subset with moderate-to-strong staining (2+ or 3+) in 40–69% of tumor cells, no significant differences in PFS or OS were observed between the groups of chemotherapy alone and the combined treatment arm with zolbetuximab [34].

The non-randomized phase II ILUSTRO trial also assessed the combination of chemotherapy with zolbetuximab in a cohort of 21 patients classified as CLDN18.2-positive, with moderate-to-strong staining (2+ or 3+) in ≥75% of tumor cells. This study demonstrated an ORR of 71.4% and a disease control rate of 100%. The pivotal phase III SPOTLIGHT and GLOW trials employed the IHC VENTANA CLDN18 (43-14A) assay kit, focusing on patients with moderate-to-strong staining (2+ or 3+) in ≥75% of tumor cells. The authors explained that the elevation of the previous cut-off from the FAST study from 70% to 75% of tumor cells was due to the use of this new automated IHC kit. The results of these studies also favored the combination of chemotherapy with zolbetuximab compared to chemotherapy alone in gastroesophageal cancer [13,14,64].

The positivity threshold can indeed exhibit variability based on the specific therapeutic approach employed. Preliminary data emerging from the phase I trial of CAR-T cell therapy targeting CLDN 18.2 (designated as CT041) in gastrointestinal tumors incorporated the IHC anti-CLDN 18.2 test (clone 14F8, a prediluted mouse monoclonal antibody developed by CARsgen) with intermediate expression defined as an intensity of 2+ or 3+ in 40–69% of stained tumor cells, while high expression was defined as an intensity of 2+ or 3+ in ≥70% of tumor cells. Among the cohort of 16 patients with high expression, the ORR was 63%. Notably, it is pertinent to underscore that even within the subset of 12 patients characterized by intermediate or low expression, a substantial response rate of 50% was distinctly evident [65].

## 6. Anti-CLDN 18.2 Therapies

### 6.1. Monoclonal Antibodies

Once CLDN18.2 became a viable therapeutic target in GC, several studies were designed by exploring anti-CLDN18.2 therapies, as summarized in Table 2. Initial trials aimed at monoclonal antibodies against CLDN18.2 and zolbetuximab (claudiximab and previously named IMAB362), which were some of the first agents to be studied. Zolbetuximab is a mouse chimeric monoclonal antibody with human IgG1 constant regions designed to bind specifically to CLDN18.2 (Figure 1) [66] with high affinity without cross-binding to other claudin family members [67,68]. Pre-clinical data demonstrate that zolbetuximab elicited antibody-dependent cell-mediated cytotoxicity in CLDN18.2-expressing GC cell lines and also induced complement-dependent cytotoxicity in CLDN18.2-positive GC cells [37]. These effects allowed an effective anti-tumor effect with low toxicity. Both preclinical and phase I studies demonstrated that zolbetuximab also delays tumor growth, prolongs survival, and attenuates metastasis development [69,70].

In 2019, Türeci and colleagues published the phase IIa MONO study [63], which enrolled 54 patients with recurrent or refractory advanced gastric of lower esophageal adenocarcinoma presenting moderate-to-strong CLDN18.2 expression in ≥50% tumor cells. Patients received zolbetuximab monotherapy every 2 weeks for five cycles. The antitumor activity of zolbetuximab was assessable in 43 patients and presented an ORR of 9% with a clinical benefit rate of 23%. In a subgroup analysis, patients with moderate-to-high CLDN18.2 expression in ≥70% of tumors cells presented a higher ORR (14%), suggesting that a high expression of CLDN18.2 might be key to improve tumor responses. Regarding toxicity, zolbetuximab presented with an 82% rate of treatment-related toxicities, of which nausea, vomiting, and fatigue were the most frequent. 

The FAST trial [34], a randomized phase II trial combining zolbetuximab with a chemotherapy backbone (EOX: epirubicin + oxaliplatin + capecitabine) versus EOX alone as first-line treatment for advanced CLDN18.2-positive gastric or gastroesophageal adenocarcinoma, included 252 patients with a moderate-to-strong CLDN18.2 expression in ≥40% of tumor cells. A benefit in PFS with EOX + zolbetuximab compared to EOX alone was achieved, with a median PFS of 7.5 months versus 5.3 months (HR: 0.44; 95% CI, 0.29–0.67; *p* < 0.0005), respectively. Also, the addition of zolbetuximab to chemotherapy was associated with a longer median OS with a reduction in the risk of death (13.0 months versus 8.3 months; HR: 0.55; 95% CI, 0.39–0.77; *p* < 0.0005). This study also demonstrated that a higher portion of patients in both trial arms presented a moderate-to-strong CLDN18.2 expression in ≥70% of tumor cells, reinforcing the idea that CLDN18.2 is a highly expressed biomarker in GC. In terms of toxicity, the majority of the adverse events presented in patients receiving the combination of zolbetuximab and EOX were nausea, vomiting, neutropenia, and anemia, and most of them were classified as grades 1–2. 

In line with the FAST trial outcomes, a recent phase II study also demonstrated interesting ORR and PFS results for the combination of zolbetuximab and chemotherapy in CLDN18.2-positive gastric or gastroesophageal adenocarcinoma [64]. The ILUSTRO phase II multicohort trial had three main study arms: zolbetuximab monotherapy in the third/later-line setting (Cohort 1A, n = 30); zolbetuximab with FOLFOX in the first-line setting (Cohort 2, n = 21); and zolbetuximab with pembrolizumab in the third/later-line setting (Cohort 3A, n = 3). This study revealed an ORR of 0% in Cohorts 1A and 3A, while an ORR of 71.4% was observed in Cohort 2. Also, the median PFS was 1.54 months in Cohort 1A, 2.96 months in Cohort 3A, and 17.8 months in Cohort 2, demonstrating that the combination of zolbetuximab and chemotherapy has promising efficacy in the first-line setting in CLDN18.2 GC, yet with no clear role for zolbetuximab in monotherapy or in combination with immunotherapy. 

The promising results of the previously discussed phase II trials prompted investigators to conduct phase III trials for CLDN18.2-positive GC patients as zolbetuximab began to surface as a promising new treatment strategy. As such, the GLOW trial [14] is a phase III double-blind randomized global study that enrolled 507 CLDN18.2-positive (≥75% of tumor cells with moderate-to-strong CLDN18 membranous staining), HER2-negative, locally advanced unresectable, or metastatic gastric/gastroesophageal junction adenocarcinoma patients. Patients were randomized to receive either zolbetuximab plus chemotherapy with CAPOX (capecitabine + oxaliplatin) or CAPOX plus placebo as first-line treatment. The authors demonstrated that the addition of zolbetuximab to chemotherapy in the first-line setting improved median PFS (8.21 versus 6.80 months; HR: 0.68; 95% CI, 0.54–0.86; *p* = 0.0007) and OS (14.39 versus 12.16 months; HR: 0.77; 95% CI 0.61–0.96; *p* = 0.011). However, it is important to note that no relevant difference in ORR was observed between the two study arms (42.5% in the zolbetuximab arm and 40.3% in the placebo arm). Regarding toxicity, nausea and vomiting were the main relevant adverse events (most observed in the first cycles) in the zolbetuximab arm, with both symptoms occurring more frequently in patients who had not previously undergone gastrectomy. 

In 2023, investigators also published the SPOTLIGHT trial [13], a randomized phase III trial also involving CLDN18.2-positive (≥75% of tumor cells with moderate-to-strong CLDN18 membranous staining), HER2-negative, locally advanced unresectable, or metastatic gastric/GEJ adenocarcinoma patients. This trial enrolled 565 individuals that were assigned to receive either zolbetuximab plus FOLFOX or placebo plus FOLFOX in the first-line setting. The SPOTLIGHT trial demonstrated an improvement in median PFS with the addition of zolbetuximab (10.6 versus 8.6 months; HR: 0.75, 95% CI 0.60–0.94; *p* = 0.0066) along with an increase in median OS (18.2 versus 15.54 months; HR: 0.75, 95% CI 0.60–0.94; *p* = 0.0053). Likewise, the addition of zolbetuximab to chemotherapy did not improve ORR. The main adverse events were also nausea, vomiting, and decreased appetite, which were higher in the first cycles of the experimental arm. Similarly to the GLOW trial, a higher proportion of patients who had not undergone gastrectomy developed vomiting when compared to patients with previous gastrectomy. 

Due to the findings of the aforementioned studies, the addition of zolbetuximab to chemotherapy in patients with CLDN18.2-positive/HER2-negative metastatic recurrent GC has become an established novel therapy in the first-line setting. As such, the FDA granted a priority review to zolbetuximab for such patients and the drug may soon become widely available in clinical practice. Along with zolbetuximab, other monoclonal antibodies are currently being studied in clinical trials [30].

### 6.2. Bispecific Antibodies

Novel agents targeting Claudin 18.2 are currently being tested in clinical trials. The most promising classes of drugs are the bispecific antibodies (bsABs) and the antibody–drug conjugates (ADCs) [30,71]. BsAB comprises two monoclonal antibodies coupled by a peptide linker, making them able to bind to two different antigens or two different epitopes on the same antigen [71,72]. 

Most bsABs are bispecific T-cell-engagers (BiTEs) and their mechanism of action rests on guiding the immune system to tumoral cells, by jointly binding CD3 on cytotoxic T lymphocytes and specific tumor antigen. Other bsAb classes include agents targeting immune checkpoints (such as PD-L1), oncogenic signaling pathways, and cytokines [73]. As for anti-CLDN18.2 bsABs and GC, the main objective of this novel option is to achieve a better tumor-cell specificity with lower toxicity [74]. Examples of promising bsABs in GC are listed below:AMG-910 confers an extended half-life and binds CD3 from T cells to CLDN18.2. This drug is being evaluated in a phase I trial (NCT04260191) including individuals with advanced GC/GEJ adenocarcinoma who had failed two or more lines of standard therapy. Accrual has been completed and results are awaited [75].Q-1802 targets both CLDN18.2 and the immune checkpoint PD-L1. It offers an antibody-dependent cytotoxicity effect on tumoral cells at the same time as blocking the PD-1 pathway, activating adaptive immunity. A phase 1 trial in solid tumors expressing CLDN18.2 is in progress [76].ASP2138 is an asymmetric 2 + 1 bsAB, with a bivalent domain for the link to CLDN18.2 and a monovalent domain for CD3 in T cells, activating a T cell-guided cytotoxicity activity directly to tumor cells [77]. A phase I trial in both GC and pancreatic cancer expressing CLND18.2 is ongoing (NCT05365581).QLS31905 also binds CD3 on the T cells, leading to its activation. In vitro and in vivo studies have demonstrated efficacy against tumor cells harboring CLDN18.2 expression. Opposed to other bsABs, QLS31905 triggers a low production of cytokine release, which may translate into a lower rate of adverse events [78]. A phase I trial is ongoing in China evaluating patients with advanced solid tumors expressing CLND18.2 (NCT05278832).ZWB67 is a novel anti-CLDN18.2 and anti-CD3 bsAB with the caveat of presenting a lower affinity for CD3 in comparison to other bsABs and, therefore, being only active in the presence of CLDN18.2-expressing cells. This mechanism is intended to reduce the immune-associated adverse events and promote more efficient on-target cytotoxicity [79]. This drug has demonstrated in vitro and in vivo efficacy, but there are no ongoing clinical trials [79].GB7004-09hu15 is a novel tetravalent bsAB (TetraBi), which specifically binds to CLDN18.2, sparing CLDN18.1. It uses a site-directed pegylation technique to generate specific tetra-scFv fragments [80,81]. It has demonstrated efficacy in vitro, but in-human studies are awaited [82].

The main limitations for the use of bsAB as target therapy for GC are splice variants and isoforms in the claudin family of proteins, their multiple transmembrane structures, and the reduced extracellular areas of antigen exposition [36]. As such, new bsAB therapies are being developed with a higher affinity and specificity [83].

### 6.3. Tri-Specific Antibodies

Preclinical data have shown that guiding T cells through CD3 is a valid strategy, but the duration of response is limited, and resistance can emerge. In order to overcome this challenge, the use of tri-specific antibodies has been tested [84]. There are two ways to use this strategy: Targeting two immune cell receptors and a specific tumor antigen: one of the targets in the T-cell is responsible for the activation of cytotoxic activity and the other is a costimulatory that prevents rapid immune exhaustion. An example is the costimulation of CD3 and CD28 [85].Targeting an immune cell receptor with two specific tumor antigens: an example is TriFlex, an antibody with a domain binding both to CD16A on natural killer cells and BCMA and CD200 on the surface of multiple myeloma cells [86].

Although some tri-specific antibodies are being tested in clinical trials, none of them involves CLDN18.2 (NCT03577028).

### 6.4. Antibody Drug Conjugates (ADCs)

ADCs are recombinant monoclonal antibodies (mAB) bound by a linker to cytotoxic agents, allowing the delivery of higher doses of cytotoxic agents with higher specificity to tumor cells and, therefore, minimizing adverse effects in normal tissues [87,88]. The cell-specific characteristic is given by binding the mAB to a specific antigen on the tumor cell surface (e.g., CLDN18.2). Upon internalization by tumor cells, the ADCs are degraded by lysosomes releasing the cytotoxic agent payload, commonly microtubule inhibitors and DNA synthesis inhibitors [30]. Some examples involving CLDN18.2 with clinical activity are as follows:EO-3021, also called SYSA-1801, is a fully human mAB also with Monomethyl Auristatin E (MMAE) as the payload, which has demonstrated activity in preclinical studies with solid tumors harboring CLDN18.2-positive cells, including tumor cells with both low and high CLDN18.2 expression [89]. Therefore, a phase 1 trial is planned (NCT 05009966).CMG901 also uses MMAE as the payload and can, additionally, stimulate antibody-and complement-dependent cytotoxic activity. A phase 1 dose-escalation trial has demonstrated safety and clinical activity in previously treated solid tumors expressing CLDN18.2 [90].

The phase I/II Chinese trial KYM901 recently evaluated the safety and efficacy of CMG901 in 113 refractory metastatic GC patients [91]. Overall, a CLDN18.2 expression of 2+ or more membrane staining in 20% or more tumor cells was found in 83% of the individuals. The ORR was 42% at the 2.2 mg/kg dose level (n = 31), 24% at the 2.6 mg/kg dose (n = 42), and 38% (n = 16) at the 3.0 mg/kg dose level (n = 16). Moreover, the 9-month OS rates at the three dose levels were 70.7%, 46.9%, and 56.3%, respectively. Only 8% of the patients discontinued the drug due to treatment-related adverse events.

RC118 is a recombinant humanized mAB with MMAE as the payload, which is also under investigation in solid tumors, but mostly in gastric and pancreatic cancers. So far, it has demonstrated a tolerable safety profile [92].TPX-4589 is also a recombinant human mAB using the payload MMAE, and it has demonstrated activity in preclinical and xenografic models of gastric and pancreatic cancers, with a higher internalization rate compared to a single mAB [93]. A phase I study is currently recruiting patients with advanced solid tumors, but for the dose expansion part of the trial, only GC and esophageal cancer will be enrolled [93].SOT102 uses PNU-159862 (a derivative product from nemorubicin) as the payload and the utility in gastrointestinal malignancies preclinical models was independent of CLDN18.2 expression levels [94], leading to a phase I/II CLAUDIO-1 trial in gastric and pancreatic cancers (NCT05525286).ATG-022 is an ADC against CLDN18.2 with MMAE as the payload. Its activity has been tested in both preclinical and xenografic models with a low expression of CLDN18.2. A phase 1 trial in advanced solid tumors (CLINCH) (NCT05718895) will be initiated in the near-future [95].

Overall, the main limitations of ADCs are antigen modifications on tumor cells (heterodimerization, mutation, downregulation) and activity dependent on the level of expression of CLDN18.2. New strategies to enhance the specificity of the antibodies in order to create an effective strategy in the scenario of low CLDN18.2 expression are being studied [96]. 

### 6.5. CAR-T Cells 

Advanced cell therapy, utilizing genetically modified T cells to express a chimeric antigen receptor (CAR), is acknowledged as a crucial approach in cancer treatment [97,98,99,100,101]. Broadly speaking, CARs are created by introducing a single-chain variable fragment (scFv) targeting a specific antigen as the extracellular domain. Moreover, signaling domains encoding modified signal transduction from the T-cell receptor, along with costimulatory molecules like CD28, 4–1BB, or OX-40, are included to enhance the activation, proliferation, and persistence of the genetically modified T cells in vivo [102].

While being hailed as a groundbreaking therapy for blood cancers, the effective utilization of CAR-T cells in treating solid tumors faces numerous hurdles. Challenges include identifying suitable tumor-specific targets and understanding the unique characteristics of the solid tumor microenvironment (TME). These difficulties stem from the hostile nature of tumors, which create unwelcoming microenvironments for the infiltration and effector functions of T cells. The solid tumor microenvironment is particularly suppressive, and cytokines play a role in sustaining this inhibitory phenotype [103].

The better understanding of the mechanisms related to the antitumor immune response has brought several opportunities of mechanistic investigation in order to overcome these barriers. In this scenario, the CLDN18.2 has emerged as a promising target for advanced cell therapy for solid tumors [104].

As the immunologic mechanism of action of monoclonal antibodies and CAR-T cells is quite different, the rationale to explore the anti-tumor activities of CLDN18.2 in advanced cell therapy platforms is very relevant. Preclinical data of humanized anti-CLDN18.2 autologous CAR-T have demonstrated antigen-specific anti-tumor effects on GC [69]. This study made the background needed to launch the first single-center clinical trial (NCT03159819) using anti-CLDN18.2 autologous CAR-T that was well tolerated in the 12 patients included, with a total ORR of 33.3% and a median PFS of 130 days [105]. In addition, an ongoing open-label multicenter, single-arm, phase I trial in which 37 patients with CLDN18.2-positive digestive system cancers were treated with CLDN18.2-targeted CAR-T cells reported the first interim analysis. All patients encountered grade 3 or more severe hematologic toxicities, but only 94.6% experienced grade 1 or 2 cytokine release syndrome (CRS), and there were no instances of grade 3 or higher CRS or neurotoxicity. The overall response rate (ORR) and disease control rate (DCR) stood at 48.6% and 73.0%, respectively. In the subgroup of patients with gastric cancer (GC), the ORR and DCR reached 57.1% and 75.0%, with a 6-month overall survival (OS) rate of 81.2% [65]. These results are in line with other anti-CLDN18.2 therapeutic strategies as very promising options for patients with GC. 

## 7. Relationship of CLDN18.2 and Other Predictive Gastric Cancer Biomarkers 

Previous work on the co-expression of CLDN18.2 and other biomarkers in advanced esophagogastric adenocarcinoma has been biased by the use of different antibodies and positivity thresholds [49,51]. However, recent data have shed light on the frequency of PD-L1 CPS, HER2, and MSI/dMMR in CLDN18.2-positive tumors as shown in Table 3. In one of the largest analyses so far, Kubota and colleagues failed to find significant differences in the expression of the PD-L1 CPS, HER2, and MSI/dMMR between CLDN18.2-positive or -negative GC [56]. In this analysis, 41.9% of CLDN18.2-positive tumors (using the currently accepted methods) had a CPS PD-L1 ≥5. In the combined analysis from the SPOTLIGHT and GLOW randomized phase III trials, 17.4% (104 out of 599) of patients had tumors with CPS PD-L1 ≥ 5 [47]. Similarly, in an Italian series of patients with advanced gastric and gastroesophageal adenocarcinomas, 17.9% of CLDN18.2-positive tumors had CPS PD-L1 expression ≥ 5 [54]. 

In the screening phase of the FAST trial, 13.8% of eligible patients had HER2-positive tumors [106]. Despite the broad CLDN18.2 eligibility criteria used in this trial (≥40% of tumor cells expressing Claudin 18.2), additional studies have demonstrated similar rates of HER2 positivity among CLDN18.2 tumors [54,56]. Again, there seems to be no difference in the expression of HER2 according to CLDN18.2 status [62]. Similar to the general population of patients with advanced esophagogastric adenocarcinoma, MSI/dMMR occurs in roughly 5% of CLDN18.2-positive tumors. Therefore, data support the concept that the expression of CLDN18.2 is not related to the expression of other biomarkers in advanced esophagogastric adenocarcinoma [107]. 

## 8. Conclusions and Future Directions

The systemic therapy of GC has dramatically changed over the past few years. The greater understanding of the molecular biology of the disease has allowed the identification of actionable molecular alterations and therapeutic targets, leading to the rising incorporation of novel agents in the armamentarium against this challenging disease. Nevertheless, the recent advances in the systemic therapy of GC did not decrease the importance of chemotherapy. The combination of fluoropyrimidines and platins remains the cornerstone of the systemic therapy of the disease, even in enriched populations for certain biomarkers. Anti-HER2 agents, immune checkpoint inhibitors, and anti-claudin 18.2 monoclonal antibodies did not replace the role of chemotherapy, but they have been added to the chemotherapy backbone.

Claudin 18.2 is a tight junction protein vital for the structure of the gastric epithelium and for the adhesion among epithelial cells. Compared to the normal and metaplastic gastric cells, claudin 18.2 may be downregulated in the gastric cancer counterparts, but it becomes more exposed to the recognition of the immune cells due to the EMT process. By attaching to the claudin 18.2 proteins, monoclonal antibodies, such as zolbetuximab, elicit an immune response through antibody-dependent cell-mediated cytotoxicity, which seems to be the main mechanism of action of anti-claudin 18.2 agents. Combined with chemotherapy doublets, zolbetuximab prolonged the PFS and OS of advanced GC patients, but did not increase the overall response rate, both in SPOTLIGHT and GLOW phase III trials. Treatment-related adverse events were manageable, yet nausea and vomiting observed in the first cycle should deserve attention. Based on these findings, the addition of zolbetuximab to FOLFOX or XELOX has become the standard of care of GC patients who present a moderate-to-strong expression of claudin 18.2 in ≥75% of the tumor cells. Data favoring immune checkpoint inhibition or anti-claudin 18.2 therapy in first-line settings have not yet been available. In the estimated 15% to 20% of the GC patients who overexpress claudin 18.2 and present PD-L1 expression CPS ≥ 5, the decision between immunotherapy and zolbetuximab in the first-line setting and the sequencing of the systemic therapy should be made on a case-by-case basis until definitive data are provided by the literature. 

Ongoing clinical trials suggest that novel agents exploring anti-claudin 18.2 as a therapeutic target will emerge in the next few years. Monoclonal antibodies with higher efficacy and lower toxicity, bispecific and trispecific antibodies, ADCs, and even CAR-T cell therapies have been developed and explored in clinical trials, and preliminary data show promising data, suggesting that anti-claudin 18.2 therapies will be ameliorated in the near-future. 

In addition to HER2, MSI, and PD-L1 status, claudin 18.2 expression will be tested in every patient with advanced GC eligible for systemic therapy. Ongoing clinical trials and agnostic therapies have also suggested that emerging biomarkers, such as fibroblast growth factor receptor 2 (FGFR2) and neurotrophic tyrosine receptor kinase (NTRK) fusions/rearrangements, as well as tumor mutational burden (TMB), will be added to the list of biomarkers to be tested in the daily clinical practice before the decision-making therapy of advanced GC. All of these findings confirm that precision medicine is a reality in the management of GC and convey a message of optimism for the patients and families affected by this challenging and heterogeneous disease.

## Figures and Tables

**Figure 1 cancers-16-00679-f001:**
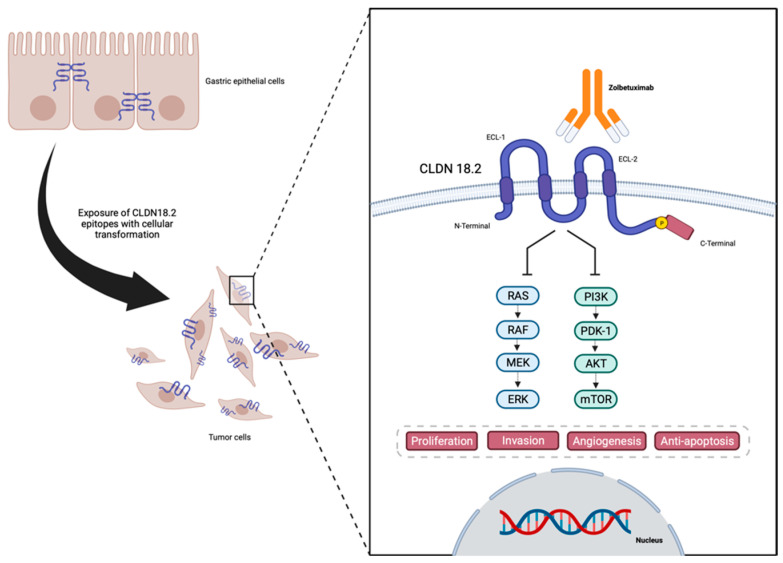
Representation of CLDN18.2 in gastric cancer cells, and structure and interaction with anti-CLDN18.2 Zolbetuximab—designed in Biorender.

**Table 1 cancers-16-00679-t001:** Studies that evaluated the prognostic role of CLDN 18.2 in GEJ/GC. OS: overall survival; * 19.8% (N = 85) gastric cancer. The color is used to highlight the statistical significance of the last for studies, as opposed to the other studies who did not present a statistical significance.

Reference	Type of Study	Country	N	Definition of CLDN 18.2 Positivity	Frequency	OS Impact
Zhu et al., 2013 [52]	Retrospective	China	329	Immunoreactivity score (IS ≥ 4)	53.2%	*p* = 0.469
Hong et al., 2020 [50]	Prospective	Republic of Korea	430 *	>5%	14.1%	*p* = 0.101
Dottermusch et al., 2019 [43]	Retrospective	Germany	481	Positive histoscore (H-score)	42.2%	*p* = 0.439
Baek et al., 2019 [44]	Retrospective	Republic of Korea	367	>50%	29.4%	*p* = 0.914
Arnold et al., 2020 [53]	Retrospective	Germany	414	Immunoreactivity score (IRS) > 8	17.1%	*p* = 0.944
Pellino et al., 2021 [54]	Retrospective	Italy	350	≥75%	33.4%	*p* = 0.926
Kayikcioglu et al., 2023 [55]	Retrospective	Turkey	65	Any positive staining	73.8%	*p* = 0.09
Kubota et al., 2023 [56]	Retrospective	Japan	408	≥75%	24%	*p* = 0.191
Resnick et al., 2005 [57]	Retrospective	USA	146	≥2+	-	*p* = 0.007
Jung et al., 2011 [58]	Retrospective	Republic of Korea	72	≥25%	44.4–73.6%	*p* = 0.046
Jun et al., 2014 [41]	Retrospective	Republic of Korea	134	≥10%	25.5–29.9%	*p* = 0.005
Kohmoto et al., 2020 [59]	Retrospective	Japan	394	High mRNA expression	18%	*p* = 0.0013

**Table 2 cancers-16-00679-t002:** Summary of relevant studies of Claudin 18.2-directed therapies. mAB: monoclonal antibody; AB: antibody; ADC: antibody–drug conjugate; CAR-T: chimeric antigen receptor T cells.

Medication Name	Type	Study Phase	Trial Number
Zolbetuximab (claudiximab/IMAB362)	mAB	III	NCT03504397, NCT03653507, NCT03505320
ABO11	mAB	I	NCT04400383
MIL93	mAB	I	NCT04671875
Osemitamab, TST001	mAB	II	NCT04495296. NCT04396821
SPX-101	mAB	I	NCT05231733
IM-102	mAB	I	NCT04735796, NCT05008445
DR30303	mAB	I	NCT05639153
ZL-1211	mAB	I	NCT05065710
TORL-2 307-MAB	mAB	I	NCT05159440
FL-301 (NBL-015)	mAB	I	NCT05153096
AMG-910	Bispecific AB	I	NCT04260191
Q-1802	Bispecific AB	I	NCT04856150
ASP2138	Bispecific AB	I	NCT05365581
QLS31905	Bispecific AB	I	NCT05278832
TPX-4589	ADC	I	NCT05001516, NCT05934331
EO-3021/SYSA-1801	ADC	I	NCT 05009966
CMG901	ADC	I	NCT04805307
RC118	ADC	I	NCT04914117, NCT05205850
SOT102	ADC	I/II	NCT05525286
ATG-022	ADC	I	NCT05718895
CAR-CLD18 T	CAR-T	-	NCT03159819

**Table 3 cancers-16-00679-t003:** Prevalence of predictive biomarkers among patients with Claudin 18.2-positive GC.

Biomarker	Kubota et al. [56] N = 98 (%)	Pellino et al. [54] N = 117 (%)
HER2	15 (15.3)	17 (14.5)
MSI-H/dMMR	5 (5.1)	15 (12.8)
PD-L1 CPS ≥ 5	39 (41.9)	21 (17.9)

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
