# Peer review of "Claudin 18.2 as a New Biomarker in Gastric Cancer—What Should We Know?"

_cancers, 2024, doi:10.3390/cancers16030679_

Round 1
Reviewer 1 Report
Comments and Suggestions for Authors
The aim of the proposed article is to provide concise insights into the potential role of CLDN18.2 as a therapeutic target and guide future clinical decisions in the management of metastatic GC. The authors make a good arguments for why research question is very important. The incidence of gastric cancer and the fact that most patients are diagnosed at advanced stage when the 5 year survival is only 6% are, certainly, reasons to investigate new methods for diagnostics and treatment follow up in gastric cancer patients. Methods are appropriate and properly conducted and the conclusion is fully supported by the data. There are no ethical concerns or other issues I believe the Editor should be aware. Therefore, I fully recommend the proposed article to be accepted for publication.
Author Response
Dear esteemed colleague,
Thank you for your revision of our manuscript and thank you for your comments.
We hope our manuscript may, in fact, enlighten readers about the applications of Claudins in gastric cancer.
Reviewer 2 Report
Comments and Suggestions for Authors
This review represents a well-made dissertation on the potential of CLDN18.2 as therapeutic target in gastric cancer. Moreover, other new derived potential therapeutic approach are considered, as well. Importantly, the article shows the therapeutic potential but also the limits of this new biological approach. It allows the reader to get a clear idea of the current state of claudin research in gastric cancer.
Author Response
Dear estemeed colleague,
Thank you for your revision of our manuscript and thank you for the kind words.
We hope our manuscript may, in fact, enlighten readers about the applications of Claudins in gastric cancer.
Reviewer 3 Report
Comments and Suggestions for Authors
In the present review, Mathias-Machado and colleagues well described the biology of CLDN and significance of applying CLDN in the clinic as a potential biomarker and therapeutic target. Here, I provide the following comments and recommendations that can be considered by the authors to improve the manuscript.
1. In lines 68-71, what do you mean by "CLDNs can be divided into two main groups according to alternative splicing of the CLDN18 gene exons"? The reference review (26) explains that the CLDNs are grouped by sequence homology.
2. In section "6. Anti-CLDN18.2 Therapies", the authors listed the therapies along with the brief information. If the authors could provide this information in a table format, it would be very helpful for the readers. Besides, if there are ongoing or completed clinical trials for a certain therapy, it will be informative if the study ID is provided together for all the therapies mentioned.
3. Although it is likely that there is no English Language issue in the text, overall proofreading of the text is recommended. For example, in Table 1., "et al" should be italicized, and the mathematical symbols and alphabet (or number) should be spaced. Likewise, please check for the rest of the text as well.
Author Response
Dear esteemed colleague,
Thank you for reviewing our manuscript and thank you for the suggestions.
In response, to your review:
1. We have rephrased the text accordingly.
2. Thank you for your suggestion, we have included a table in the manuscript where the relevant therapies have been summarized and grouped by therapeutic class with the study Phase and Clinical Trial number.
3. Thank you . We have made the necessary modifications.